# Utility of the Levonorgestrel-Releasing Intrauterine System in the Treatment of Abnormal Uterine Bleeding and Dysmenorrhea: A Narrative Review

**DOI:** 10.3390/jcm11195836

**Published:** 2022-10-01

**Authors:** Paola Bianchi, Sun-Wei Guo, Marwan Habiba, Giuseppe Benagiano

**Affiliations:** 1Department of Medico-Surgical Sciences and Translational Medicine, Sant’Andrea Hospital, Sapienza, University of Rome, 00161 Rome, Italy; 2Shanghai OB/GYN Hospital, Fudan University, Shanghai 200011, China; 3Shanghai Key Laboratory of Female Reproductive Endocrine-Related Diseases, Fudan University, Shanghai 200011, China; 4Department of Health Sciences, University Hospitals of Leicester, University of Leicester, Leicester LE1 7RH, UK; 5Faculty of Medicine and Dentistry, Sapienza, University of Rome, 00161 Rome, Italy

**Keywords:** abnormal uterine bleeding, amenorrhea, dysmenorrhea, levonorgestrel-releasing intrauterine system, Progestasert

## Abstract

Introduction: We undertook a literature review of the use of levonorgestrel-releasing intrauterine devices when utilized for heavy menstrual bleeding and/or dysmenorrhea. Methods: A narrative review of articles in the Scopus and Medline databases was conducted. Results: A number of options exist for the management of both abnormal uterine bleeding (AUB) and dysmenorrhea, and evidence is accumulating that the insertion of a levonorgestrel-releasing intrauterine system (LNG-IUS) represents a useful option for their long-term treatment. The idea of using a progestogen released in utero was initially conceived to achieve long-term contraception, but it was quickly found that these systems could be utilized for a number of therapeutic applications. The first device to be made commercially available, Progestasert, was withdrawn from the market because, in the event of contraceptive failure, it caused a disproportionate percentage of extrauterine pregnancies. On the other hand, the LNG-IUS continues to be successfully utilized in its various variants, releasing 20, 13, or 8 μg/day. These devices have a respective duration of action of 7 (possibly 8), 5, and 3 years, and there exist versions of frameless systems affixed to the myometrium of the uterine fundus. In the present review, following a brief description of the major causes of AUB and dysmenorrhea, the molecular bases for the use of the LNG-IUS are summarized. This is followed by a compendium of its use in AUB and dysmenorrhea, concluding that the insertion of the system improves the quality of life, reduces menstrual blood loss better than other medical therapies, and decreases the extent of dysmenorrhea and pelvic pain. In addition, there is no evidence of a significant difference in these outcomes when the use of the LNG-IUS was compared with improvements offered by endometrial ablation or hysterectomy. Possibly, the most important mechanism of action of the system consists of its ability to induce amenorrhea, which effectively eliminates heavy bleeding and dysmenorrhea. However, no method is ideal for every woman, and, in the case of the LNG-IUS, younger age and severe dysmenorrhea seem to be associated with a higher risk of discontinuation. Conclusion: The higher-dose LNG-IUS is a useful tool for HMB and dysmenorrhea in women of all ages. The low cost and ease of use make the LNG-IUS an attractive option, especially when contraception is also desired.

## 1. Introduction

In their recent comprehensive review of the regulation of menstruation, Critchley et al. [1] reminded us that the menstruating endometrium is a physiological example of an injured or “wounded” surface requiring rapid repair each month. They also pointed out that abnormal uterine bleeding (AUB) imposes a massive burden on one in four women of reproductive age. The estimates by Davis and Sparzaks [2] indicate that up to 1/3 of all women will experience some form of AUB in their lifetime, most commonly around menarche and perimenopause.

In addition, a comprehensive review of longitudinal, case–control, and cross-sectional studies found a wide difference of between 16% and 91% in the prevalence of dysmenorrhea in women of reproductive age and that 2–29% of them suffered severe pain. Factors linked to reduced risk are older age, higher parity, and the use of oral contraceptives. High stress and a family history increase the risk of dysmenorrhea. There is inconclusive evidence about the role of smoking, diet, obesity, depression, or abuse [3]. Dysmenorrhea affects women across the globe [3,4].

In recognition of the magnitude of the problem, the plethora of possible and often coexisting causes and the inconsistency in the nomenclature used to describe AUB, the International Federation of Gynecologists and Obstetricians (FIGO) [5] proposed a comprehensive classification system aimed at facilitating both clinical care and research coined the “PALM-COEIN” (polyp; adenomyosis; leiomyoma; malignancy and hyperplasia; coagulopathy; ovulatory dysfunction; endometrial; iatrogenic; and not yet classified). It systematically classifies AUB into groups with distinctively different structural and functional causes. Recently, FIGO also proposed a new classification for ovulatory disorders [6].

An illustration of the PALM-COEIN system is provided in Figure 1.

A number of options exist for the management of both AUB and dysmenorrhea. In 2016, Bradley and Gueye [7] summarized the process of dealing with the two conditions as follows: Take a thorough history and perform a physical examination and proper imaging studies; once all significant structural causes are excluded, proceed with medical management. In the presence of an acute episode of AUB with a normal uterus, parenteral estrogen, multidose combined oral contraceptives (COCs), multidose progestin-only regimen, and tranexamic acid are all indicated. If heavy bleeding is a common feature, then a levonorgestrel-releasing intrauterine system (LNG-IUS), COCs, continuous oral progestins, and tranexamic acid represent efficacious methods.

With regard to dysmenorrhea, a recent review by McKenna and Fogleman [8] recommended starting with the menstrual history and a pregnancy test for sexually active patients, followed by nonsteroidal anti-inflammatory drugs (NSAIDs), COCs, and an LNG-IUS, all considered first-line medical options.

In clinical practice, AUB and dysmenorrhea often coexist [9], and therefore, from a practical point of view, the same treatment can be applied in the presence of both symptoms. It should be noted that both conditions represent important non-contraceptive indications of pharmacological agents that were first utilized for fertility regulation. Such applications have been known for decades and were systematically listed in 2010 by the American College of Obstetricians and Gynecologists (ACOG), which provided a comprehensive list of such benefits [10,11]. Specifically, they affirmed that “*Hormonal contraception should be considered for women with menorrhagia who may desire future pregnancies… Data on the effects of the levonorgestrel intrauterine system on dysmenorrhea are limited, but because the device reduces or eliminates menstruation for many women, these benefits seem consistent with the mechanism of action… The levonorgestrel intrauterine system reduces blood loss by up to 86 percent after three months, and by up to 97 percent after 12 months*”.

In the last few years, new drugs, such as dienogest, selective progesterone receptor modulators (SPRMs), and GnRH antagonists, have become available for the management of HMB and dysmenorrhea due to uterine fibroids and adenomyosis. However, the use of SPRMs is restricted by European Medicine Agency (EMA) warnings, and GnRH antagonists are still very pricy. Given the importance of the benefits of the LNG-IUS for these two indications and its low cost and excellent safety profile, we decided to critically review the molecular, biological, and clinical information on the use of the LNG-IUS to treat AUB and dysmenorrhea.

## 2. Methods

This narrative review is based on articles identified through a search of Scopus and Medline databases until March 2022 using the headings HMB, menorrhagia, dysmenorrhea, and AUB, together with levonorgestrel (or norgestrel) intrauterine device or system and progestogen intrauterine device or system. We searched for major publications, including systematic reviews, meta-analyses, randomized controlled trials (RCTs), and consensus concerning the utility of the LNG-IUS in the treatment of abnormal uterine bleeding and dysmenorrhea. The reference lists of identified articles were used to search for any additional relevant references. Articles published in languages other than English were excluded. This article consists of a narrative review to synthesize the findings in the literature, which was conducted following the Scale for Assessment of Narrative Review Articles (SANRA).

## 3. Definition and Management of Abnormal Uterine Bleeding

As mentioned, there is a comprehensive definition and classification system for all types of AUB, “PALM-COEIN” [5]. More specifically, Warner et al. [12,13] defined the most common form, menorrhagia (a term increasingly replaced by the expression: “heavy menstrual bleeding”, HMB), in terms of a statistical “abnormality”, namely, as a blood loss of >80 mL per cycle, as first investigated by Hallberg et al. more than 50 years ago [14]. The use of the term “menorrhagia” should be discouraged, as it has been applied loosely, notwithstanding the fact that it does have a precise definition.

In routine clinical practice, measurements are seldom carried out. Warner et al. [12] found that, in their cohort, only 34% (95% Confidence Interval (CI) = 28–40%) of women had a volume of blood loss >80 mL, associated with the subjective heaviness of the period. Women reported a range of problems associated with their periods [14]: dysmenorrhea (37.5%), mood changes (35.7%), and changes over time in the amount (volume) of menses (33.8%). Interestingly, the absolute volume was only fourth among their complaints (31.2%). They concluded that the 80 mL criterion for menorrhagia is of limited clinical usefulness, since it does not predict the presence of problems or the iron status. Therefore, it does not help in guiding management [13].

In contemporary times, HMB has been defined as excessive menstrual blood loss (MBL) that disrupts the physical, social, emotional, and/or material quality of life [15]. Over the years, a number of guidelines have been produced to manage HMB (e.g., [15,16,17]), and excellent reviews on the subject have been published [18,19,20,21,22].

Of particular relevance here is the Cochrane review of the use of progestogen-releasing intrauterine systems by Bofill Rodriguez et al. [23] that included 25 randomized clinical trials (RCTs), totaling 2511 women, comparing the LNG-IUS versus other medical therapies. They concluded that the LNG-IUS may improve HMB, lowering the MBL measured through the alkaline hematin method (mean difference (MD) 66.91 mL; 95% CI = 42.61–91.20) and the Pictorial Bleeding Assessment Chart (PBAC) (mean = 55.05; 95% CI = 27.83–82.28). The use of the LNG-IUS may have a favorable effect on women’s satisfaction for up to one year (RR = 1.28; 95% CI = 1.01–1.63) and result in a slightly higher quality of life.

The definition of heavy menstrual bleeding has evolved over time. The current interpretation places it within the broad classification of abnormal uterine bleeding or AUB.

## 4. Definition and Management of Dysmenorrhea

Dysmenorrhea is defined by the presence of pain during menstruation. Pain can start before the onset of bleeding. It represents the most common menstrual symptom among adolescent girls and young women. If painful menstruation occurs in the absence of pelvic pathology, this is referred to as primary dysmenorrhea [24,25]. Dysmenorrhea is a common, often debilitating condition affecting between 45 and 95% of menstruating women, and it is often poorly treated or even disregarded in spite of its impact [24].

Following a critical review of published information, Iacovides et al. [24] concluded that women with primary dysmenorrhea acquire a greater sensitivity to pain both within and outside the period of menstruation. This leads to a significantly reduced quality of life, poorer mood, and poorer sleep quality during menstruation. In their view, the first-line treatment consists of the use of NSAIDs.

Guidelines for the management of dysmenorrhea have been produced by the UK National Institute for Health and Care Excellence (NICE) [25]; in addition, ACOG issued an opinion on how to manage dysmenorrhea associated with endometriosis in adolescent girls [26].

The management of endometriosis-associated pelvic pain and dysmenorrhea has been specifically investigated by Bahamondes and his group [27,28,29]. First, they investigated the use of biomarkers by measuring the serum levels of cancer antigen-125 (CA-125), cluster of differentiation 23 (CD23), and the endometrial nerve fiber (ENF) density before and after the insertion of the LNG-IUS or the etonogestrel (ENG) contraceptive implant. They observed that both systems significantly reduced the concentration of serum-soluble CD23 and the ENF density (*p* < 0.001); interestingly, CA-125 was significantly reduced only among users of the ENG implant (*p* < 0.05). Unfortunately, no correlation was observed between the reduction in biomarkers and improvements in visual analog scale (VAS) pain and dysmenorrhea scores. They then carried out a randomized comparative clinical trial of the ENG implant and the LNG-IUS and found that both systems significantly improved the mean VAS score. In addition, health-related quality of life improved significantly, with the most common bleeding pattern for the LNG-IUS at 180 days of follow-up being infrequent bleeding and spotting, as shown in Figure 2.

Dysmenorrhea is a very common presentation and is linked to a reduced quality of life. The use of the LNG-IUS improves the quality of life and pain scores, but there is no correlation with pain biomarkers.

## 5. The Progesterone-Releasing Intrauterine System

Contrary to a commonly held view, the concept of using a progestogen released in utero for contraception dates back more than fifty years: the first experimental device, reported in 1968, utilized medroxyprogesterone acetate (MPA) [30], but the initial system was abandoned because all of the devices were expelled within a month [31]. More successful were the attempts to first use a Lippes loop and then use the so-called Tatum-T as carriers for progesterone [32,33]. A progesterone-releasing device called Progestasert was marketed in the mid-1980s by Alza Corporation with a stated duration of one year [34]. It had a 1.3% failure rate, but quickly became the therapeutic device of choice for women with dysmenorrhea or heavy menstrual flow [35]. Unfortunately, epidemiological data evidenced that, in the event of contraceptive failure, the device caused a disproportionate percentage of extrauterine pregnancies [36], and, not withstanding its secondary therapeutic action, it was withdrawn from the market.

In spite of its short commercial life, a number of interesting observations of the biological effects of Progestasert were published. In particular, the mechanism of action was evaluated in the 1970s and 1980s, showing that, with time, the intrauterine administration of progesterone causes the increasing suppression of the proliferative activity of the endometrium, a clear pre-decidual stromal reaction, poorly developed glands, and a decrease in vascularity, accompanied by an increase in the proportion of small vessels with defects [37,38,39,40]. However, in a controlled study, the overall concentration of these small blood vessels was significantly lower than in normal subjects, with an average blood vessel density of 2.39 (range, 13.0 to 3.71) compared to 3.92 for the controls (range, 3.33 to 4.68) (*p* = 0.01) [40]. This led the authors to suggest that this phenomenon may account for significantly less uterine blood loss in Progestasert users. The proportion of defective vessels in controls ranged between 0% and 24%, and in the Progestasert group varied from 7.1% to 64%, with average percentages of 13.4% and 35.0%, respectively (*p* = 0.001).

In conclusion, the early stages of development of progestogen-releasing devices utilized MPA; this was followed by the use of progesterone microcrystals. The development of both types of devices was discontinued because of high expulsion rates in the case of the MPA device and because of the increased risk of ectopic pregnancy with the progesterone device.

## 6. Levonorgestrel-Releasing Intrauterine Systems

In 1975, an international group under the auspices of the Population Council [41] began testing a polylactate film releasing levonorgestrel (LNG) (in those days, called D-norgestrel), with the rates of the diffusion of the steroid and of the hydrolysis of the polymer suggesting that it could be utilized for intrauterine contraception. As a carrier, they used a copper-releasing T device stripped of the copper wire. The initial work showed promise for developing an intrauterine device capable of slowly releasing LNG over a period of 2–3 years.

In the following year, the same group [42] carried out a Phase I clinical trial in six women volunteers using a device composed of a Tatum-T-IUD in which the carrier for LNG was a sleeve of poly-dimethyl-siloxane (Silastic). The device was tested in six healthy women, and plasma concentrations of d-norgestrel (today renamed levonorgestrel), estradiol, and progesterone were determined for periods of 112–114 days. They observed plasma values of the steroid corresponding to a daily release of 50 pg, sufficient to suppress ovulation and produce irregular but reduced bleeding.

Since the early experience, it has become clear that the system has a positive effect on the amount of blood loss, as documented by Nilsson [43] (See Figure 3).

Endometrial changes were also evaluated with devices releasing either 20 or 40 μg/day, observing a uniform suppression of the endometrium with glandular atrophy and the decidualization of the stroma [44].

By 1986, information had become available proving that the device’s effectiveness as a contraceptive lasted 5 years [45], and in 2021, the US Food and Drug Administration (FDA) approved its use for contraception for up 8 years [46,47] following the publication of evidence of its long-term effectiveness [48]. When used for HMB, the license is for 5 years.

As mentioned, a reduction in MBL has been observed since the very early clinical experience, and this led to the utilization of the device in subjects with HMB. The first application dealt with the prevention and treatment of iron-deficiency anemia [49], followed by the publication of the results of the use of the LNG-IUS for 1 year in a small cohort of 20 women with menorrhagia (>80 mL blood loss per menstrual period). There was a significant reduction in blood loss, reaching 97% at 1 year, when, in addition, there was also a significant increase (*p* < 0.001) in serum ferritin [50]. Soon afterward, it was proven that the reduction in MBL achieved by the LNG-IUS was significantly greater than that recorded with flurbiprofen (*p* < 0.001) and tranexamic acid (*p* < 0.01) [51].

By the turn of the millennium, besides menorrhagia and iron-deficiency anemia, several additional therapeutic indications were identified; these included: protection of the endometrium in estrogen replacement therapy; dysmenorrhea; endometrial hyperplasia; ovarian endometrioma; adenomyosis; and endometrial protection in women treated with tamoxifen during breast cancer treatment [52,53]. In a comprehensive review published in 2006 [54], additional indications were mentioned: fibroid-related menorrhagia, pelvic-inflammatory disease (not proven), and endometrial cancer. A number of reviews confirmed these indications [55,56,57,58,59]. However, caution should be exercised in women with malignancy, and whilst it may reduce endometriosis-related pain, it does not resolve endometriomas.

An interesting point was raised by Cristobal et al. [60], who prospectively evaluated the quality of life (QoL) of LNG-IUS users over the first-year post-insertion using the so-called “Spanish society of contraception quality-of-life” questionnaire, developed to specifically assess the impact of contraceptive methods on the health-related (HR) QoL. This index rose from a mean of 46.3 (±17.3) at baseline to 72.2 (±14.8) at 12 months (*p* < 0.001). Overall, 94.6% of the participating women claimed to have received additional benefits besides contraception.

In conclusion, as stressed by Gemzell-Danielsson et al. [61], the availability of the LNG-IUS has had a considerable impact on women’s wellbeing far beyond the reduction in the need for abortion and surgical sterilization. It reduced the number of hysterectomies carried out for HMB and, in the context of the COVID-19 pandemic, provided a treatment option for women with some gynecological issues without organic pathology, minimizing their exposure to the hospital environment and waiting times for surgical appointments.

The successful development of the LNG-IUS has had a considerable impact beyond its use as an effective contraceptive. The LNG-IUS has proven to be an effective alternative to hysterectomy for HMB.

### Type of Devices Available

The characteristics of levonorgestrel-releasing intrauterine systems (LNG-IUSs) available at present are summarized in Table 1.

Basically, four types of systems have been marketed: the first three utilize a T-shaped IUD, the vertical arm of which contains different quantities of LNG:(1)**LNG-IUS-20**: Total content of 52.5 mg; LNG is initially released at a rate of 20 μg/day, and—as already mentioned—in 2021, the US FDA approved its use for contraception for up to 8 years [47]. It is licensed for HMB for up to 5 years.(2)**LNG-IUS 12**: Total content of 19.5 mg; LNG is initially released at a rate of 13 μg/day and has a duration of action of 5 years [62]. The device is not licensed for HMB.(3)**LNG-IUS 8**: Total content of 13.5 mg; LNG is initially released at a rate of 8 μg/day and has a duration of action of 3 years [63]. The device is not licensed for HMB.

The fourth system consists of a 3.5 cm-long coaxial fibrous LNG delivery system with an approximate release of 14 μg/day. The calculated duration of release from the device is at least 3 years [64].

In China, a similar frameless LNG-IUS affixed to the myometrium has been tested in patients with adenomyosis and HMB [65] in a comparative trial of 60-month duration. They observed, at 1-year, significant changes in the pretreatment severity of dysmenorrhea, menstrual volume, uterine volume, and hemoglobin level in each group (*p* < 0.01 in all groups).

The introduction of frameless devices is claimed to overcome cramps as a symptom in some users of framed devices. It may also be suitable in the presence of fibroids that distort the uterine cavity [66].

As shown in Table 1, the original LNG-IUS, commercially known as Mirena©, is currently marketed along with generic versions. Ilyin et al. [67] compared these generic systems with the reference product Mirena© in subjects with HMB. They randomly assigned Donasert or Mirena to a total of 312 subjects and observed mean absolute changes in MBL of −130 (71.8) mL and −127 (67.3) mL at 6 months in the Donasert and Mirena groups, respectively; the non-inferiority of Donasert was confirmed (*p*-value < 0.0001).

Recently, the WHO recommended that the classification system used for intrauterine contraception be made clearer. For instance, the WHO recommended that the type of hormone be specified (e.g., levonorgestrel-releasing IUD) [68]. The recommendation of the Society of Family Planning in the USA recommended that the dose release also be specified and that the word “releasing” be dropped. Thus, available devices would be referred to as Levonorgestrel 13.5 mg, 19.5 mg, or 52 mg IUDs [69]. It is important here to reiterate that the release rate constantly decreases with the duration of hormonal IUD use.

## 7. Causes of HMB and Dysmenorrhea and the Molecular Basis for the Use of LNG-IUS

As seen from the FIGO classification, the causes of AUB can be attributable to PALM-COEIN [70]. HMB is probably the most common form of AUB. The FIGO classification system lists the contributing causes of AUB (except for those instances that are not yet classified), but the underlying mechanisms are incompletely understood.

HMB has been linked to systemic causes, including hematological or thyroid disorders, as well as local uterine causes, such as fibroids or adenomyosis. However, a large proportion of women with HMB have no systemic or structural abnormalities, and the underlying mechanisms reside in the endometrium.

Following progesterone withdrawal in the late secretory phase in the absence of pregnancy, the endometrial expression of cyclooxygenase-2 (*COX-2*), a gene coding for the rate-limiting prostanoid-synthesizing enzyme, is elevated, resulting in subsequent increased levels of prostaglandins (PGs), particularly PGE_2_ and PGF_2α_ [71,72]. The induction of endometrial PGE_2_ signaling, along with the endometrial generation of the vasoconstrictor PGF_2α_, leads to local hypoxia in the upper functional layer of the endometrium. This is marked by the activation of hypoxia-inducible factor 1α (HIF-1α), the master regulator of cellular hypoxia, which, as such, establishes an endometrial microenvironment that is conducive to tissue repair and angiogenesis [73,74]. However, any disruption of hypoxia signaling or PGE_2_ signaling in the endometrium will impair endometrial repair, causing HMB [1,75].

However, despite this knowledge, the more detailed mechanisms underlying adenomyosis-, endometrial-polyp-, and uterine-fibroid-induced HMB are largely unknown. For example, submucosal fibroids are more likely to be associated with HMB [76], but why this is so is less clear. However, a recent study indicates that the increased lesional fibrosis in adenomyosis, especially when located near the endometrium, can disrupt hypoxia and PGE_2_ signaling, causing HMB [77]. This is because increased lesional fibrosis creates a microenvironment conducive to further fibrogenesis in neighboring tissues and a stiffened extracellular matrix (ECM), which, in turn, accentuates the actions of transforming growth factor β1 (TGF-β1) and promotes myofibroblast activation [78,79,80]. As a result, lesional fibrosis will propagate to the neighboring endometrial–myometrial interface (EMI) and then to the eutopic endometrium. As the matrix stiffness increases and interferes with multiple steps of PGE_2_ synthesis, including the suppression of prostaglandin E synthases (PTGESs) that specifically convert the precursor PGH_2_ to PGE_2_ [81], PGE_2_ production and signaling subside [82,83,84,85]. Increased fibrosis may also enhance PGE_2_ degradation, as seen in the increased expression of the PGE_2_-degrading enzyme 15-hydroxy-prostaglandin dehydrogenase (PGDH) in the fibrotic lung [85]. As a result, there will be a subsequent suppression of PGE_2_,s as well as HIF-1α signaling in adenomyotic lesions and their EMI and endometrium, with ensuing impaired endometrial repair and, ultimately, HMB. Indeed, endometriotic stromal cells cultured on a stiff matrix display reduced expression of COX-2, prostaglandin E2 receptor 2 (EP2), and EP4 [83]. It is suspected that a similar mechanism may also be at work for submucosal fibroids [77].

The molecular mechanisms underlying dysmenorrhea are thought to be more complex. Conceivably, however, uterine hyperactivity, hyperinnervation within the uterus, the release of inflammatory cytokines and pain mediators by fibroids or adenomyotic lesions, and central sensitization may collectively contribute to dysmenorrhea.

The high local concentration of the progestin steadily released by the LNG-IUS suppresses the cellular proliferation of endometrial cells [86], thus inhibiting endometrial growth and resulting in a thinner endometrium. In addition, the pseudo-pregnancy (leading to pseudo-decidualization) conferred by the LNG-IUS results in no endometrial injury and thus no or substantially less bleeding. Moreover, the progestin released by the LNG-IUS changes the prostaglandin ratio by stimulating arachidonic acid formation in the endometrium [87] in favor of endometrial repair and thus less MBL.

The progestins contained in the LNG-IUS can also suppress uterine contractility [88,89,90], removing one important irritant that contributes to dysmenorrhea. Progestins can also activate progesterone receptors in the myometrium, antagonizing nuclear factor kappa-light-chain-enhancer of activated B cells (NF-κB) and thus reducing inflammation [91], removing another contributor to dysmenorrhea. As an added bonus for less or absent menstrual bleeding, there is less platelet aggregation and reduced or attenuated thrombin activation, which may further hinder uterine contractions [92,93].

Progesterone is central in endometrial remodeling and acts on both the epithelial and stromal compartments. Since a progestin is the major active agent within LNG-IUS, and since the action of progesterone/progestins is mediated through progesterone receptors (PRs), the ultimate efficacy of the LNG-IUS hinges critically on the expression levels of PRs, particularly PR isoform B (PR-B) in the endometrium and myometrium. In both adenomyotic and endometriotic lesions [94,95] and in endometrial hyperplasia and endometrial cancer [96,97,98], as well as in endometrial polyps [99], PR expression is all reduced. Hence, the LNG-IUS-released progestin, as a ligand, is likely to induce PR expression and exerts its therapeutic effect.

However, in many cases, the induction of PR expression can be difficult, if not impossible. For example, PR-B has been reported to be hypermethylated [100,101] and presumably silenced in endometriotic and adenomyotic lesions. In adenomyotic lesions in particular, the response to progestin treatment hinges critically on the mutational status of Kras [102]. Since it is unlikely that the methylation status or the DNA sequence is changed by progestin treatment, progestins may not be effective in containing these lesions when genetic/epigenetic aberrations are present.

In addition, for uterine leiomyoma, progesterone/progestins may actually stimulate cellular proliferation and fibrogenesis, effectively accelerating the disease progression [103,104,105], although evidence is not univocal: On the one hand, high doses of the progestin MPA are capable of promoting leiomyoma growth [104]. On the other hand, there is evidence that long-term progestogen administration reduces the occurrence of fibroids [106,107,108], and MPA administration prevented any significant re-growth following the treatment of women with symptomatic fibroids with the short-acting Gn-RH superagonist analog Buserelin [109]. Finally, it has been shown that an add-back therapy of progestins with GnRH agonists reverses the efficacy of GnRH agonists in reducing fibroid size [110,111]. That is, progesterone/progestins may actually facilitate the progression of uterine leiomyoma. Still, treatment can be effective in reducing HMB.

Then, why is the LNG-IUS still useful in treating HMB and dysmenorrhea caused by endometriosis, adenomyosis, or leiomyoma? The LNG-IUS is effective in reducing menstrual blood loss. Among women using the LNG-IUS for contraception, amenorrhea or spotting was reported in approximately 50% of users during the last 90 days of the first year after insertion; with time, 80% experienced amenorrhea, spotting, or light bleeding only [112]. After 60 months of follow-up, 96.2% of women with HMB experienced amenorrhea, oligomenorrhea, or spotting, whilst 95.8% and 93.3% of women with adenomyosis or fibroids reported improvements [113]. This may stop or reduce the release of inflammatory and pain mediators associated with menstruation and, as such, dysmenorrhea. As Brosens pointed out in 1997 [114], endometriosis is characterized by recurrent ectopic bleeding, and, as such, the suppression of recurrent menstrual bleeding is in itself efficient in the treatment of symptomatic endometriosis. For other causes of HMB or dysmenorrhea, the rationale remains true as well.

In conclusion, although the molecular mechanisms are incompletely understood, the LNG-IUS proved effective in primary and some cases of secondary dysmenorrhea and in reducing HMB, including in women with fibroids.

## 8. Use in Abnormal Uterine Bleeding

By the end of the previous millennium, several investigators had tested the use of the LNG-IUS in the treatment of AUB. In 1997, two reports provided evidence of a positive effect in subjects with heavy bleeding: Fedele et al. [115] inserted the device in 25 women with recurrent adenomyosis-related menorrhagia. Of the 23 women who completed 12 months of treatment, 2 had become amenorrheic, 3 were oligomenorrheic, 2 reported spotting, and 16 had regular periods. The authors speculated that the IUS produced decidualization and, subsequently, marked hypotrophy of the eutopic endometrium. Barrington and Bowen-Simpkins [116] inserted the LNG-IUS in 50 women awaiting surgery and evaluated menstrual loss using a pictorial chart, a full blood count, and the measurement of ferritin. By nine months post-insertion, bleeding was reduced to acceptable levels in 41 cases, with 4 subjects developing amenorrhea. These results were subsequently confirmed in larger cohorts [117,118].

A series of investigations tested the efficacy of the LNG-IUS in women harboring leiomyomas, with interesting results [119,120,121,122,123,124]. With one exception [112], they all concluded that the insertion of the system was associated with a profound reduction in menstrual blood loss and a significant reduction in uterine volume. At the same time, the LNG-IUS does not significantly reduce the volume of leiomyomas. One investigation [124] found a significant reduction in visual bleeding scores and spotting with an increase in amenorrhea and uterine pulsatility index scores, whereas another [16] observed a drop in the PBAC from a mean value of 231.7 to 17.6 at 12 months. In addition, the duration of menstrual bleeding diminished significantly (*p* < 0.001), with a good satisfaction rate in 89% of all cases. It can be concluded that the LNG-IUS is effective in controlling both menorrhagia and/or frequent irregular uterine bleeding related to the presence of myomas, though with no significant effect on the myoma size.

In an RCT that included women with idiopathic HMB, the LNG-IUS was more effective than cyclical MPA given for 10 days starting from day 16 of the cycle. The LNG-IUS was linked to a greater absolute and percentage reduction in menstrual blood loss and a higher likelihood of success [125].

Using the “frameless” LNG-IUS, Wildemeersch and Schacht [126] evaluated a small group of 14 subjects with the device in place for more than 1 year and 29 subjects with the device for 6 months or more. All women reported greatly reduced blood loss, although there were no cases of amenorrhea; the reduction in bleeding was substantial after 1 month of treatment and further decreased over the next months to remain stable thereafter.

An interesting investigation aimed at determining whether the size of the uterine cavity is a determining factor in causing bleeding and pain in nulligravid women using the LNG-IUS [127], and the authors concluded that a small uterine cavity size is in fact beneficial, causing an increase in the rate of amenorrhea and decreasing the presence of pain. Recently, Beelen et al. [128], in a cohort of 201 women with HMB, observed a discontinuation rate of 46% at 24 months, with age <45 (adjusted RR = 1.51; 95% CI = 1.10–2.09; *p* = 0.012) and severe dysmenorrhea (adjusted RR = 1.36; 95% CI = 1.01–1.82; *p* = 0.041) being associated with a higher risk of discontinuation. Anticipatory counseling can reduce premature removal of the device.

A recent investigation enrolled a cohort of 1714 subjects and followed them for 5 years, with 495 participants finishing the entire period. As shown in Figure 4, the rate of amenorrhea among these women increased with time, and at the end of the five years, it almost reached 42% [129].

It is important to mention that Cho et al. [130], on the one hand, reported a significant reduction in pain scores, menstrual blood loss, and uterine volume in women with adenomyosis after one year of LNG-IUS insertion; on the other, they found a significant increase in all parameters at 36 months of use compared to 12 months, suggesting a reduction in efficacy in the longer term.

This may be due to a reduction over time in the amount of LNG released by the system. According to the FDA, the initial release of the steroid is 20 μg/day, but this is reduced to around 10 ug/day over time. In terms of plasma levels, from the initial 150–200 pg/mL during the first weeks, there is a reduction to 180 ± 66 pg/mL, 192 ± 140 pg/mL, and 159 ± 59 pg/mL at 12, 24, and 60 months, respectively [46].

To conclude, a 2015 systematic review [131] found that the use of the LNG-IUS improved the quality of life and reduced menstrual blood loss better than standard medical therapy. It also found no evidence of a significant difference in these outcomes when the use of the system was compared with the improvements offered by endometrial ablation or hysterectomy.

In conclusion, the effectiveness of the LNG-IUS in women with HMB is well documented, but anticipatory counseling is important for avoiding premature removal. The LNG-IUS that releases 20 ug/day is required to control HMB. Its use is associated with endometrial pseudo-decidualization.

### 8.1. Use in Women with Bleeding Disorders

The link between abnormal bleeding and hematological disorders has been recognized for some time [132]. Attention to this phenomenon was drawn by a group of hematologists at the London Free Hospital around the turn of the millennium [133,134]. They reported that menorrhagia affected 74% of patients with von Willebrand disease (vWD), 57% of carriers of hemophilia A or B, 59% with Factor XI deficiency (FXID), and 60% with Factor VII deficiency, compared with 29% of control subjects.

At that time, tranexamic acid (TxA) and desmopressin acetate were utilized to control menorrhagia in these women; then, in 2004, the London Group, for the first time, proposed using the LNG-IUS [135]. They carried out a prospective pilot study in 16 women with treatment-resistant menorrhagia caused by the following inherited bleeding disorders (IBDs): vWD (n = 13), FXID (n = 2), and Hermansky–Pudlak syndrome, a genetic disorder characterized by, i.a., platelet dysfunction (n = 1). Subjects were followed up for nine months, and all women reported improvements in their periods, with lower pictorial chart scores; 56% became amenorrhoeic.

Soon after, another investigation explored the use of the LNG-IUS during anticoagulant therapy through the distribution of a questionnaire, with 17 responders [136]. The study reported that, following insertion, there was a reduction in bleeding in 10 women, and amenorrhea occurred in 4 cases. There was no change in one patient and an increase in bleeding in the last two. By 2007, Kadir and Chi [137] had identified four studies suggesting that the LNG-IUS is a viable and safe option for the management of menorrhagia in women with IBDs. Additional cases were published by Lukes et al. [138].

The results of the long-term follow-up (mean duration of 33 months; range: 14–103) of 26 subjects with IBDs were presented by Chi et al. [139]: they found that while using the LNG-IUS, the median PBAC score decreased from 255 (range, 134–683) to 35 (range, 0–89); the median hemoglobin (Hb) concentrations rose from 11.2 to 13.2 g/dL, and the QoL scores improved significantly, with the median rising from 26 to 52 (*p* < 0.01).

Two important complications of the LNG-IUS in these subjects, malposition and expulsion, were evaluated by Rimmer et al. [140], starting from the accepted expulsion rates of between 5 and 10% in normal subjects. The majority of subjects studied (12/20) were affected by type 1 VWD; their median age at the time of insertion was 31 years (range 18–43 years). There were three LNG-IUS expulsions and two episodes of device malposition resulting in removal (25.0%). An additional five women had their devices removed prematurely. From this study, it seems that a significant proportion of women with an IBD had the device removed due to malposition or expulsion. Obviously, the small number of subjects (a common feature of all investigations in IBD women) suggests caution in assessing these figures.

A small study from China reported on the successful use of the LNG-IUS in two adolescents with Glanzmann’s Thrombasthenia, an IBD caused by defects in the platelet membrane glycoproteins IIb/IIIA. Insertion of the system produced a significant reduction in menstrual blood loss [141]. Another Chinese report [142] mentioned the case of a woman affected by primary myelofibrosis, a rare condition secondary to progressive pancytopenia; she suffered from HMB in spite of using a COC, and the administration of norethisterone yielded an unsatisfactory hemostatic effect. At this stage, an LNG-IUS was inserted, and at the 5-month follow-up, the patient reported a lower menstruation bleeding volume.

With time, more substantial cohorts became available: Brull et al. [143] reviewed the charts of 117 women attending a clinic for reasons related to IBDs who used the LNG-IUS. Ninety-nine of them had a history of thrombosis (with 71.7% using oral anticoagulants), and eighteen had coagulopathies. Approximately two-thirds of these subjects reported amenorrhea or oligomenorrhea at 12, 24, and 54 months of follow-up. No difference was observed between the groups with a history of thrombosis or coagulopathy, or between users and non-users of oral anticoagulants. A second study [144] followed 20 subjects with IBDs wearing the LNG-IUS; they observed a progressive reduction in the median PBAC score over time, becoming highly significant at 12 months (*p* < 0.001). At that time, 70% of the subjects were amenorrheic. Finally, there was a substantial (*p* < 0.001) improvement in eight parameters of quality of life (QoL), as well as in the levels of hemoglobin, ferritin, and serum iron.

At present, guidelines are available for the gynecologic and obstetric management of women with vWD derived from three systematic reviews of the literature comparing the effects of desmopressin, hormonal therapy, and TxA on HMB [145]. When comparing desmopressin to COCs, a single observational study suggested that both are equally effective at alleviating symptoms (RR = 0.90; 95% CI = 0.66–1.23). With regard to the LNG-IUS, the review found that, unfortunately, case series could not be included in the meta-analysis. At any rate, they found very low certainty for the comparative effectiveness of the LNG-IUS with other therapies in the control of HMB, the duration of menstruation, health-related quality of life, anemia, absence from necessary activities, complications, and adverse effects.

### 8.2. The Effect of the LNG-IUS on the Endometrium

Locally released LNG has a profound effect on the endometrium: as already mentioned, back in 1978, changes in the endometrium were evaluated with devices releasing either 20 or 40 μg/day, observing a uniform suppression of the endometrium [44]. Under the influence of locally delivered LNG, there is a thinning of the mucosa to about 1–3 mm. The stroma becomes edematous and features pseudo-decidualization of the cells. The endometrial glands are scarce and atrophic with no mitosis, and there is leukocytic infiltration [146].

### 8.3. Comparison of Blood Loss with Cu-IUD

In an early investigation, Nilsson [43] quantified and compared menstrual blood loss with the LNG-IUS and a Nova-T-copper device (Cu-IUD) in a group of 19 volunteers and calculated, over three consecutive menstrual periods, a blood loss of 20.7 ± 6.0 mL for the former and 72.5 ± 6.2 mL for the latter. In addition, over time, the amount of blood loss significantly decreased during the use of the LNG-IUS.

Some 35 years later, Lowe and Prata [147] carried out a systematic review of changes in Hb and ferritin and compared the effects of the LNG-IUS with those of the Cu-IUD. They found 14 investigations involving the copper device in nonanemic subjects and 4 in anemic subjects; these were compared to the results of reviewing 6 trials of the levonorgestrel system. The meta-analyses for hemoglobin changes showed significant decreases for bearers of Cu-IUDs and an increase for the LNG-IUS. In the case of ferritin, the levels followed the same pattern. Interestingly, decreases in mean hemoglobin values in bearers of the copper device were not sufficient to induce anemia in previously nonanemic women.

An investigation in India [148] compared hemoglobin and ferritin in two groups of 50 subjects and found that the insertion of the LNG-IUS reduced the number of menstrual bleeding days and increased both hemoglobin and serum ferritin levels. Specifically, in users of the LNG-IUS, at one year, besides Hb and ferritin, there were also increases in red blood cells, packed cell volume, mean corpuscular volume, mean corpuscular hemoglobin, and mean corpuscular hemoglobin concentration, whereas the values of all of these parameters decreased in the Cu-IUD group. These observations may be especially important for women in developing countries, where a decrease in blood loss may improve iron-deficient conditions.

## 9. Use in Dysmenorrhea

The early investigation by Barrington and Bowen-Simpkins [116] observed that 56% of their cohort of 41 women noticed considerable improvements in, or the outright cure of their premenstrual syndrome symptoms, whereas 80% reported a reduction in the severity of dysmenorrhea (SDys). Ten years later, Bahamondes et al. [149] carried out a review of the medical literature on the effect of the LNG-IUS on pelvic pain and dysmenorrhea and found that all trials reported improvements in both parameters. One study with a follow-up of three years, while finding improvement in pelvic pain at 12 months of use, reported no further improvement after that.

Over the years, additional information has become available: Yucel et al. [150], in subjects with endometriomas, found a decrease in the VAS score of pain intensity from 6.13 to 2.88 and of dyspareunia from 6.04 to 2.61 at 12 months. Yoo et al. [151] focused their attention on the hysterectomy rates and the risk factors for hysterectomy during the first 2 years of use of the LNG-IUS in perimenopausal women complaining of either menorrhagia or dysmenorrhea. They found that out of 192 women over 40 years of age, 26 (13.5%) had to be hysterectomized because of the failure of the system; however, the severity of pain before treatment was not a factor for referral for surgery. On the other hand, surgery was correlated with the pain score at three months.

In a longitudinal population study, Lindh and Milsom [152] addressed a vital question: whether intrauterine contraception influences the prevalence of SDys. They compared SDys in the same woman using a Cu-IUD, the LNG-IUS, or a COC with other methods of contraception or no contraception. Intriguingly, they found that SDys was unchanged in the same women when using a Cu-IUD compared with using other methods (condom, barrier methods, natural family planning, coitus interruptus, and sterilization) or no method of contraception. LNG-IUS and COC use were associated with reduced SDys compared with other methods/no method. Finally, SDys decreased between the ages of 19 and 44 years.

Kelekci et al. [153] compared the effect of the LNG-IUS and a Cu-IUD on menstruation and dysmenorrhea in women with and without adenomyosis and found that the LNG-IUS significantly improved SDys, as well as the characteristics of menstrual bleeding and Hb levels in subjects with adenomyosis. They also observed that acne was significantly increased in LNG-IUS bearers. Over a period of 3 years, Sheng et al. [154] followed 94 women with adenomyosis and moderate or severe dysmenorrhea; they observed that the dysmenorrhea VAS score dropped continuously and significantly from a score of 77.9 ± 14.7 to 11.8 ± 17.9 at 36 months (*p* < 0.001). This drop was accompanied by a significant decrease in the uterine volume, and the overall satisfaction rate for the treatment was 72.5%.

Information is also available on the “frameless” LNG-IUS (FibroPlant) developed by a Belgian Group. In a preliminary trial, Wildemeersch et al. [155] tested the new device in 22 subjects, of whom 8 had primary dysmenorrhea and 10 had secondary dysmenorrhea. All women reported substantially reduced pain, or no pain at all. The overall performance of this system was summarized in 2017, and in subjects with dysmenorrhea, the device was found to be as effective as the classic one [65].

In conclusion, LNG-IUS and frameless devices have been linked to an improvement in dysmenorrhea.

## 10. Specific Issues

For decades, it has been maintained that intrauterine contraception is contraindicated in adolescents and young and nulliparous women, in spite of evidence to the contrary [156,157].

### 10.1. Barriers to Use

Buhling et al. [158], in eight European countries plus Canada, evaluated knowledge, attitudes, and beliefs in over 1000 healthcare providers vis-à-vis intrauterine contraception (IUC), concluding that the predominant concerns were nulliparity and pelvic inflammatory disease (PID) for women in general and insertion difficulty and pain, PID, and infertility for nulliparous women; this in spite of the fact that the World Health Organization Medical Eligibility Criteria for IUC lists “nulliparity” under category 2 (benefits outweigh risks) [159]. Focusing on Canada, Hauck and Costescu [160] concluded that to overcome misperceptions about the use of IUC, it is necessary to educate care providers, women, and policy makers.

### 10.2. Use in Young Women

Back in 2010, a short-term comparative trial [161] showed that at 6 months in adolescent girls (14–18 years old), continuation rates tended to be greater with the LNG-IUS than with the Cu-IUD. In the same year, another investigation [162] reported on 48 adolescents (mean age 15.3 years) bearing the LNG-IUS for over 8 years, mostly because of menorrhagia and dysmenorrhea resistant to oral treatment. In the vast majority (93.4%), there was a significant improvement in their menstrual symptoms.

Clear advantages exist in treatment with the LNG-IUS in adolescents with HMB, dysmenorrhea, and pelvic pain/endometriosis, and, indeed, good results have been reported in young women with AUB, dysmenorrhea, and pelvic pain related to endometriosis [163].

### 10.3. Safety, Side Effects, and Reasons for Failure

A description of the adverse effects of the LNG-IUS is beyond the scope of this review; the reader may refer to review articles on the subject (e.g., [110,164,165,166]).

On the other hand, it is perfectly relevant to briefly summarize existing views on the pathophysiological mechanism through which side effects manifest themselves.

Jimenez et al. [167] tried to quantify subendometrial microvascularization and uterine artery blood flow in subjects bearing either the LNG-IUS or the T-Cu 380A. They observed increased subendometrial blood flow in subjects with severe dysmenorrhea and/or AUB after controlling for the device type, age, and parity. However, the pulsatility and resistance indices were not different in these women, leading to the conclusion that measuring these parameters could become a prognostic factor to better evaluate the risk of developing dysmenorrhea and/or bleeding after device insertion.

An attempt to identify factors associated with the failure of the LNG-IUS in controlling HMB was made by Beelen et al. [128], who used data from a cohort of women aged 34 years and older, without intracavitary pathology and without plans for future children, treated with the device and suffering from AUB. A multivariable analysis showed younger age (age below 45) (adjusted RR = 1.51; 95% CI = 1.10–2.09; *p* = 0.012) and severe dysmenorrhea (adjusted RR = 1.36; 95% CI = 1.01–1.82; *p* = 0.041) to be associated with a higher risk of discontinuation.

## 11. Conclusions

Over the last quarter of a century, evidence has been accumulating showing that the LNG-IUS, in all of its variants, represents a useful tool to combat both AUB and dysmenorrhea in women of all ages. Almost all published trials have reported improvements for both conditions. Perhaps the most important mechanism of action of the LNG-IUS is attributable to its ability to induce amenorrhea, which effectively eliminates HMB and dysmenorrhea. While new generations of therapeutics are now becoming available, the low cost and ease of use make the LNG-IUS very attractive, especially when contraception is also desired. Still, for large uteri, the LNG-IUS may be less effective, and there has been limited level I data on its efficacy in treating endometrial-polyp-induced HMB. Regardless, the LNG-IUS is and is expected to remain, in the foreseeable future, a viable option for treating AUB and dysmenorrhea.

## Figures and Tables

**Figure 1 jcm-11-05836-f001:**
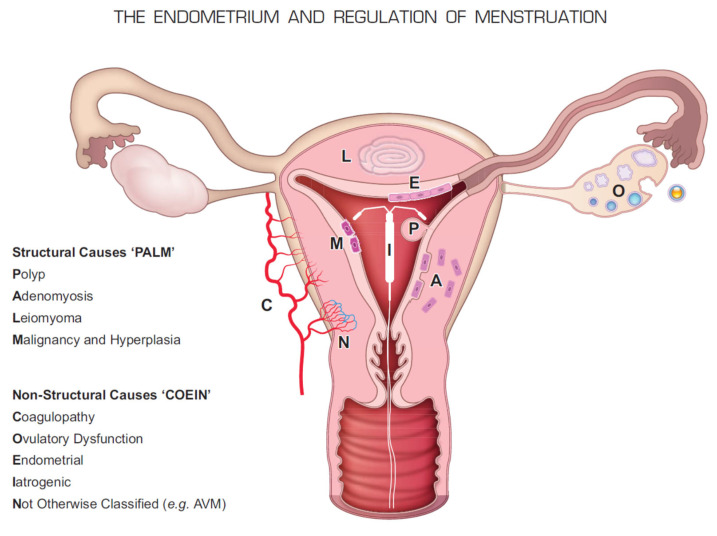
Illustration of the PALM-COEN classification system for abnormal uterine bleeding. From Critchley et al. (2020) [1] (reproduced with permission).

**Figure 2 jcm-11-05836-f002:**
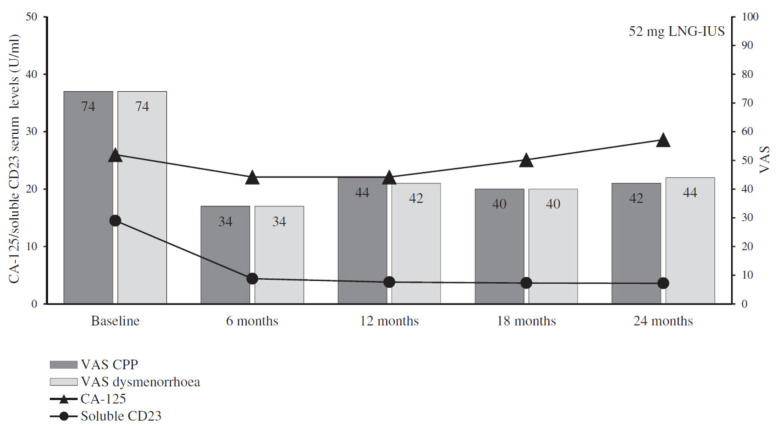
Serum levels of cancer antigen-125 (CA-125), cluster of differentiation 23 (CD23), visual analog scale (VAS), and chronic pelvic pain (CPP) over two years of treatment with the LNG-IUS. From: Margatho et al. (2020) [28] (reproduced with permission).

**Figure 3 jcm-11-05836-f003:**
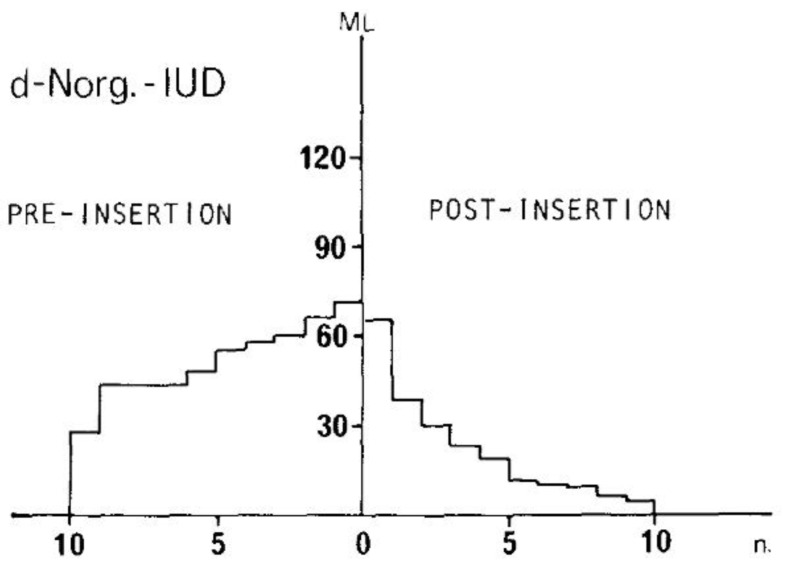
Duration and intensity of menstrual blood loss before and during 3 cycles after insertion of an experimental levonorgestrel-releasing device. From: Nilsson (1977) [42] (reproduced with permission).

**Figure 4 jcm-11-05836-f004:**
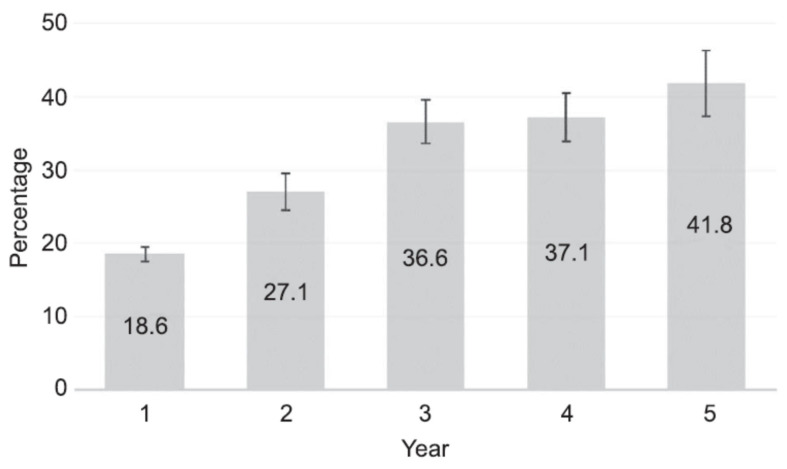
Rates of amenorrhea (defined as no bleeding or spotting during the preceding 90 days) over time in women bearing the LNG-IUS. From: Teal et al. (2019) [105] (reproduced with permission).

**Table 1 jcm-11-05836-t001:** Various types of devices available on the market.

**LNG-IUS 20:** Total content: 52.5 mg; daily LNG release: 20 μg; FDA-approved duration of use: 8 years for contraception, 5 years for HMB. Available as: ***Mirena***^®^ (Bayer, Germany); ***Levosert^®^*** (Gedeon Richter) (known as ***Donasert^®^*** in Russia and France; ***Benilexa^®^*** in Italy; and ***Liletta***^®^ in the United States).
**LNG-IUS 12**: Total content: 19.5 mg; daily LNG release: 13μg; duration of use: 5 years (for contraception). Available as: ***Kayleena***^®^ (Bayer).
**LNG-IUS 8**: Total content: 13.5 mg; daily LNG release: 8μg; duration of use: 3 years (for contraception). Available as: ***Jaydess*® *(Bayer)*;** ***Skyla*® *(Bayer)*.**
**Frameless 3.5 cm-long coaxial fibrous LNG-IUS**: Daily LNG release: 14μg; duration of use: 3 years. Available as: ***Fibroplant*® *(Contrel Research)*;** ***GyneFix (Tianjin Medic, Medical Equipment Co., Ltd.)*.**

## Data Availability

No unpublished data from the Authors were reported.

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
