# Peer review of "Utility of the Levonorgestrel-Releasing Intrauterine System in the Treatment of Abnormal Uterine Bleeding and Dysmenorrhea: A Narrative Review"

_jcm, 2022, doi:10.3390/jcm11195836_

Round 1

Reviewer 1 Report

Dear authors,

Thank you for this great review. I feel that your paper can contribute to scientific knowledge, but some adjustments are required. Please inform in the title that your paper is a narrative review. Also, include a small paragraph informing how you chose which papers would be included (keywords and databases searched).

Lines 49-51. Prevalence and incidence are different; cross-sectional studies provide us prevalence data (% of people with AUB); cohorts are the best way to obtain incidence (new cases among susceptible population). Please rewrite this paragraph with accurate information on each.

Lines 52-53. Why did you choose to describe only Ethiopan women? Please include some extra references on other countries, to make it broader.

Line 102. FIGO advices against using terms such as "menorrhagia", please avoid it or make it clearer that this definition is no longer recommended.

Line 124. Perhaps you intended to say "Use of the LNG-IUS may have a reasonable effect on women's satisfaction"?

Line 188. Please verify the information "levonorgestrel (LNG) (in those days called D-norgestrel)", since it seems plausible that LNG and D-norgestrel are isomers, but not the same.

Line 212. FDA has just approved 8 years use.

Lines 226-227. One must be careful when saying that ovarian endometrioma and breast cancer are indications for LNG-IUS; although this device is excellent for alleviating endometriosis-associated pain, it is not used to reduce endometriomas. We also need to be careful concerning breast cancer, since WHO eligibility criteria make it clear that any hormonal contraception is contraindicated for women with current breast cancer.

Line 258. FDA has just approved 8 years use.

Line 267. Please mind the nomenclature, "Kyleena".

Line 303. Please remove the comma: "Ilyin et al. [65] have compared..."

Line 312. Please be careful while mentioning "spotting" since this word means "any bloody vaginal discharge that is not sufficient to require protection" (Belsey EM, Machin D, d'Arcangues C. The analysis of vaginal bleeding patterns induced by fertility regulating methods. World Health Organization Special Programme of Research, Development and Research Training in Human Reproduction. Contraception. 1986;34(3):253-60).

Line 356. How do you feel about changing "pseudo pregnancy" by "pseudo decidualization"?

Lines 386-387. Reference 100 does not show that "high doses of the progestin MPA are capable of promoting leiomyoma growth", as you said. Please review if the reference is correct.

Line 388. Reference 104 shows that add-back therapy might interfere with uterine volume reduction; still, it can be used due to improvement of uterine bleeding and hemoglobin levels. Please make it clearer to avoid that readers falsely consider this treatment useless.

Line 393. Not only amenorrhea, but achieving favorable bleeding patterns is desirable. LNG-IUD 52 mg has good therapeutic properties to reduce AUB and uterine volume (Magalhaes J, Ferreira-Filho ES, Soares-Junior JM, Baracat EC. Uterine volume, menstrual patterns, and contraceptive outcomes in users of the levonorgestrel-releasing intrauterine system: A cohort study with a five-year follow-up. Eur J Obstet Gynecol Reprod Biol. 2022;276:56-62).

Line 434. Please emphasize that most women "discontinued treatment within six months after LNG-IUS insertion", that is, premature removals, probably due to an insufficient anticipatory counseling.

Line 479. What does "IBD" stands for?

Line 518. Please notice that we cannot say that "desmopressin may be less effective at alleviating symptoms (RR=0.90; 95% CI=0.66-1.23)" if 95% CI includes the unit.

Concerning AUB, authors could consider including some comments on non-structural causes, including the recently published paper by Munro et al. (2022): The FIGO Ovulatory Disorders Classification System. Besides, an interesting RCT was not mentioned: Kaunitz AM, Bissonnette F, Monteiro I, Lukkari-Lax E, Muysers C, Jensen JT. Levonorgestrel-releasing intrauterine system or medroxyprogesterone for heavy menstrual bleeding: a randomized controlled trial. Obstet Gynecol. 2010;116(3):625-32. For structural causes, it would be interesting to mention Magalhaes J, Ferreira-Filho ES, Soares-Junior JM, Baracat EC. Uterine volume, menstrual patterns, and contraceptive outcomes in users of the levonorgestrel-releasing intrauterine system: A cohort study with a five-year follow-up. Eur J Obstet Gynecol Reprod Biol. 2022;276:56-62. It is also necessary to follow the Society of Family Planning Committee statement on IUD nomenclature (Creinin M, Kohn JE, Tang JH, Serna TB; Society of Family Planning Clinical Affairs Committee. Contraception. 2022;106:1-2.), replacing IUS by IUD and informing LNG dosages of the included studies.

Wish you all the best and good luck!

Author Response

REVIEWER 1

Thank you for this great review. I feel that your paper can contribute to scientific knowledge, but some adjustments are required. Please inform in the title that your paper is a narrative review. Also, include a small paragraph informing how you chose which papers would be included (keywords and databases searched).

Reply: Many thanks for your comments. We will add “narrative review” to the Title and we will add a paragraph on how we selected the articles to cite.

  1. Lines 49-51. Prevalence and incidence are different; cross-sectional studies provide us prevalence data (% of people with AUB); cohorts are the best way to obtain incidence (new cases among susceptible population). Please rewrite this paragraph with accurate information on each.

Reply: You are, of course, correct. Have corrected the paragraph

  1. Lines 52-53. Why did you choose to describe only Ethiopian women? Please include some extra references on other countries, to make it broader.

Reply: We have clarified that the intention is to demonstrate the dysmenorrhea affects women across the globe. (Hong Ju et al., 2014)

  1. Line 102. FIGO advices against using terms such as "menorrhagia", please avoid it or make it clearer that this definition is no longer recommended.

Reply: The present text says: “As mentioned, there is a comprehensive definition and classification system for all types of AUB, the “PALM-COEIN” [5]. More specifically, Warner et al [11,12] defined the most common form, menorrhagia (a term increasingly replaced by the expression: “heavy menstrual bleeding”, HMB), in terms of statistical "abnormality", namely as a blood loss of >80 mL per cycle, as first investigated by Hallberg et al more than 50 years ago [13]”. We thought that the expression “a term increasingly replaced by the expression: ‘heavy menstrual bleeding’, HMB” was sufficient. Following your advice, we have rephrased it with “a term increasingly replaced by the expression: ‘heavy menstrual bleeding’, HMB, since the use of the term menorrhagia has been discouraged by FIGO”. Having done this, we wish to mention that the term menorrhagia has a definition, although people misuse it.

  1. Line 124. Perhaps you intended to say "Use of the LNG-IUS may have a reasonable effect on women's satisfaction"?

Reply: You are correct. We have added the word “favorable” to the text.

Line 188. Please verify the information "levonorgestrel (LNG) (in those days called D-norgestrel)", since it seems plausible that LNG and D-norgestrel are isomers, but not the same.

Reply: This is an old story. There are two isomers of norgestrel and one is not active. Originally, the active isomer was labelled by chemists as the ‘D isomer’. Subsequently, chemists decided that the active form was the ‘levo’ (‘l-isomer) one. This change by chemists did create confusion among endocrinologists and gynecologists and one of us, at the time working in WHO, suggested to call the active form “levonorgestrel” to avoid any confusion. Therefore, ‘D-norgestrel’ and ‘levo-norgestrel’ are the same substance and the term “levonorgestrel” is currently the most commonly used.

  1. Line 212. FDA has just approved 8 years use.

Reply: Indeed. It happened after we submitted our text. We have amended the text. And inserted a new reference (FDA Extends Mirena IUD for Eight Years of Use. August 19, 2022. https://www.formularywatch.com/view/fda-extends-mirena-iud-for-eight-years-of-use. We will amend the text, also pointing out that the license for treating women with HMB is 5 years.

  1. Lines 226-227. One must be careful when saying that ovarian endometrioma and breast cancer are indications for LNG-IUS; although this device is excellent for alleviating endometriosis-associated pain, it is not used to reduce endometriomas. We also need to be careful concerning breast cancer, since WHO eligibility criteria make it clear that any hormonal contraception is contraindicated for women with current breast cancer.

Reply: We realize that the text may lead to the conclusion that we agree with all of the indications proposed. To avoid this, we substituted the verb “identified”, with “proposed”. We will also insert the statement: “However, caution should be exercised in women with malignancy and we pointed out the issue about endometriomas”.

  1. Line 258. FDA has just approved 8 years use.

Reply: See above.

  1. Line 267. Please mind the nomenclature, "Kyleena".

Reply: Sorry!

  1. Line 303. Please remove the comma: "Ilyin et al. [65] have compared..."

Reply: Done

  1. Line 312. Please be careful while mentioning "spotting" since this word means "any bloody vaginal discharge that is not sufficient to require protection" (Belsey EM, Machin D, d'Arcangues C. The analysis of vaginal bleeding patterns induced by fertility regulating methods. World Health Organization Special Programme of Research, Development and Research Training in Human Reproduction. Contraception. 1986;34(3):253-60).

Reply: We are aware of this publication. We have modified the sentence.

  1. Line 356. How do you feel about changing "pseudo pregnancy" by "pseudo decidualization"?

Reply: We used the term as mentioned in the article in question. We have now added the new term in brackets.

  1. Lines 386-387. Reference 100 does not show that "high doses of the progestin MPA are capable of promoting leiomyoma growth", as you said. Please review if the reference is correct.

Reply: We have replaced Reference 100 with: Kim JJ, Sefton EC. The role of progesterone signaling in the pathogenesis of uterine leiomyoma. Mol Cell Endocrinol. 2012;358:223-31. doi: 10.1016/j.mce.2011.05.044. We have also expanded the text and explained that there is evidence that long term progestogen administration reduces the occurrence of fibroids (Venkatachalam S, Bagratee JS, Moodley J. Medical management of uterine fibroids with medroxyprogesterone acetate (Depo Provera): a pilot study. J Obstet Gynaecol. 2004; 24:798-800 – Johnson N, Fletcher H, Reid M. Depo medroxyprogesterone acetate (DMPA) therapy for uterine myomata prior to surgery. Int J Gynaecol Obstet 2004; 85:174-6 – Harmon QE, Baird DD. Use of depot medroxyprogesterone acetate and prevalent leiomyoma in young African American women. Hum Reprod. 2015;30:1499-504. Finally, we have mentioned that following treatment of women harboring symptomatic leiomyomata with the short-acting Gn-RH superagonist analog Buserelin, the administration of MPA prevented any significant re-growth (Benagiano G, Morini A, Aleandri V, Piccinno F, Primiero FM, Abbondante G, Elkind-Hirsch K. Int J Gynaecol Obstet. 1990;33:333-43. doi: 10.1016/0020-7292(90)90520-u).

  1. Line 388. Reference 104 shows that add-back therapy might interfere with uterine volume reduction; still, it can be used due to improvement of uterine bleeding and hemoglobin levels. Please make it clearer to avoid that readers falsely consider this treatment useless.

Reply: Agreed. We have modified the text, bearing in mind that it has been shown that an add-back therapy of progestins with GnRH agonists reverses the efficacy of GnRH agonists in reducing fibroid size.

  1. Line 393. Not only amenorrhea, but achieving favorable bleeding patterns is desirable. LNG-IUD 52 mg has good therapeutic properties to reduce AUB and uterine volume (Magalha Belsey es J, Ferreira-Filho ES, Soares-Junior JM, Baracat EC. Uterine volume, menstrual patterns, and contraceptive outcomes in users of the levonorgestrel-releasing intrauterine system: A cohort study with a five-year follow-up. Eur J Obstet Gynecol Reprod Biol. 2022; 276:56-62).

Reply: We have modified the text. The online-version of this article was published in July and therefore was not in our original list. We will, of course, add this reference.

  1. Line 434. Please emphasize that most women "discontinued treatment within six months after LNG-IUS insertion", that is, premature removals, probably due to an insufficient anticipatory counseling.

Reply: We will do as you suggested.

  1. Line 479. What does "IBD" stands for?

Reply: Here it means “inherited bleeding disorders”. It is mentioned for the first time on line 469 and there we have now spelled it out.

  1. Line 518. Please notice that we cannot say that "desmopressin may be less effective at alleviating symptoms (RR=0.90; 95% CI=0.66-1.23)" if 95% CI includes the unit.

Reply: you are correct. We have amended the text.

  1. Concerning AUB, authors could consider including some comments on non-structural causes, including the recently published paper by Munro et al. (2022): The FIGO Ovulatory Disorders Classification System. Besides, an interesting RCT was not mentioned: Kaunitz AM, Bissonnette F, Monteiro I, Lukkari-Lax E, Muysers C, Jensen JT. Levonorgestrel-releasing intrauterine system or medroxyprogesterone for heavy menstrual bleeding: a randomized controlled trial. Obstet Gynecol. 2010;116(3):625-32. For structural causes, it would be interesting to mention Magalhaes J, Ferreira-Filho ES, Soares-Junior JM, Baracat EC. Uterine volume, menstrual patterns, and contraceptive outcomes in users of the levonorgestrel-releasing intrauterine system: A cohort study with a five-year follow-up. Eur J Obstet Gynecol Reprod Biol. 2022;276:56-62. It is also necessary to follow the Society of Family Planning Committee statement on IUD nomenclature (Creinin M, Kohn JE, Tang JH, Serna TB; Society of Family Planning Clinical Affairs Committee. Contraception. 2022;106:1-2), replacing IUS by IUD and informing LNG dosages of the included studies.

Reply: We have added text to mention the references suggested by the reviewer.

Reviewer 2 Report

Although it is an interesting topic there some points need improvement

1. Abstract should have the forms "introduction/methods/results/conclusion

2. Authors mention that it is a  review, but they did not make a flow chart about how the did the literature search

3. There are too many informations presented, that's why it is difficult to read. I think authors should focus only in one topic. Either to molecular basis, or to compare the forms of intrauterine systems according to the literature

4. Many of the information provided, are already known and extensively explained previously. Authors should focus on new information and explain the differences

Author Response

REVIEWER 2

Although it is an interesting topic there some points need improvement.

  1. Abstract should have the forms "introduction/methods/results/conclusion

Reply: the reviewer is correct. We have modified the Abstract accordingly.

  1. Authors mention that it is a review, but they did not make a flow chart about how they did the literature search

Reply: Reviewer 1 has also mentioned this. We have added a section to the text.

  1. There are too many informations presented, that's why it is difficult to read. I think authors should focus only in one topic. Either to molecular basis, or to compare the forms of intrauterine systems according to the literature

Reply: We beg to disagree. If we did what has been suggested, we would be accused of being incomplete. Our viewpoint seems shared also by Reviewers #1 and #3, who welcomed the text in its present format.

  1. Many of the information provided, are already known and extensively explained previously. Authors should focus on new information and explain the differences

Reply: here again we beg to disagree. By definition, a review presents an up-to- date of information already published. Its scope is to place all this information in one text, so that the reader finds everything he/she may wish to know.

Reviewer 3 Report

I red with interest the manuscript " UTILITY OF THE LEVONORGESTREL-RELEASING INTRAUTERINE SYSTEM IN THE TREATMENT OF ABNORMAL UTERINE BLEEDING AND DYSMENORRHEA", which provided a comprehensive overview of the LNG-IUD history  and its the most important mechanism of action...The article describes some details of the primary dysmenorrhea' s aetiology, either. 

The authors cited published systematic reviews and others observational studies relating effectiveness of LNG-IUD and HMB, and dysmenorrhea. Among specific issues I would recommend to add  available data on use of the LNG-IUD in a case of congenital heart deceases on coagulants with dysmenorrhea and HMB.  

Not all of the available generic names of LNG-IUD were  mentioned  (for example, "Fleree", Bayer) 

Author Response

REVIEWER 3

I read with interest the manuscript " UTILITY OF THE LEVONORGESTREL-RELEASING INTRAUTERINE SYSTEM IN THE TREATMENT OF ABNORMAL UTERINE BLEEDING AND DYSMENORRHEA", which provided a comprehensive overview of the LNG-IUD history and it’s the most important mechanism of action.

The article describes some details of the primary dysmenorrhea's aetiology, either.

Reply: We wish to thank the reviewer for her/his positive outlook on our work.

The authors cited published systematic reviews and others observational studies relating effectiveness of LNG-IUD and HMB, and dysmenorrhea. Among specific issues I would recommend to add available data on use of the LNG-IUD in a case of congenital heart deceases on coagulants with dysmenorrhea and HMB.

Reply: As recommended by the reviewer, we searched for “Use of the LNG-IUS in cases of congenital heart diseases on coagulants with dysmenorrhea and heavy bleeding”. We found no specific reference to this issue; the two articles we retrieved (Khajali Z, Ziaei S, Maleki M. Menstrual Disturbances in Women with Congenital Heart Diseases. Res Cardiovasc Med. 2016;5: e32512. doi: 10.5812/cardiovascmed.32512 – Roos-Hesselink JW, Cornette J, Sliwa K, Pieper PG, Veldtman GR, Mark R. Johnson MR. Contraception and cardiovascular disease. Eur Heart J. 2015;36:1728-34. doi: 10.1093/eurheartj/ehv141), do not mention the LNG-IUS.

Not all of the available generic names of LNG-IUD were mentioned (for example, "Fleree", Bayer)

Reply: This is the 13.5mg (LNG-IUS 8) name used in Estonia. We suggest that, for the purpose of this review, we do not need to list all trade names across the world. However, if felt useful, we are ready to do so.

Round 2

Reviewer 2 Report

Authors made a good effort and improved their manuscript. However, there still some points need further improvement

1. If the authors would like to keep all the information provided in their study, it would be better if they make a table with all the issues. This could make their manuscript easier to read

2. In their manuscript there is no discussion

3. Flow chart means a diagramm presenting the literature search, articles excluded and articles included and how many aricles finally they included
